# The Assisi Think Tank Meeting Breast Large Database for Standardized Data Collection in Breast Cancer—ATTM.BLADE

**DOI:** 10.3390/jpm11020143

**Published:** 2021-02-19

**Authors:** Fabio Marazzi, Valeria Masiello, Carlotta Masciocchi, Mara Merluzzi, Simonetta Saldi, Paolo Belli, Luca Boldrini, Nikola Dino Capocchiano, Alba Di Leone, Stefano Magno, Elisa Meldolesi, Francesca Moschella, Antonino Mulé, Daniela Smaniotto, Daniela Andreina Terribile, Luca Tagliaferri, Gianluca Franceschini, Maria Antonietta Gambacorta, Riccardo Masetti, Vincenzo Valentini, Philip M. P. Poortmans, Cynthia Aristei

**Affiliations:** 1Fondazione Policlinico Universitario “A. Gemelli” IRCCS, UOC di Radioterapia Oncologica, Dipartimento di Diagnostica per Immagini, Radioterapia Oncologica ed Ematologia, 00168 Roma, Italy; fabio.marazzi@policlinicogemelli.it (F.M.); luca.boldrini@policlinicogemelli.it (L.B.); elisa.meldolesi@guest.policlinicogemelli.it (E.M.); daniela.smaniotto.rt@gmail.com (D.S.); luca.tagliaferri@policlinicogemelli.it (L.T.); mariaantonietta.gambacorta@policlinicogemelli.it (M.A.G.); vincenzo.valentini@policlinicogemelli.it (V.V.); 2Fondazione Policlinico Universitario “A. Gemelli” IRCCS, 00168 Roma, Italy; carlotta.masciocchi@guest.policlinicogemelli.it; 3Radiation Oncology Section, Department of Medicine and Surgery, University of Perugia, 06123 Perugia, Italy; maramerluzzi@libero.it; 4Radiation Oncology Section, Perugia General Hospital, 06123 Perugia, Italy; saldisimonetta@gmail.com; 5Fondazione Policlinico Universitario “A. Gemelli” IRCCS, UOC di Diagnostica per Immagini, Dipartimento di Diagnostica per Immagini, Radioterapia Oncologica ed Ematologia, 00168 Roma, Italy; paolo.belli@policlinicogemelli.it; 6Istituto di Radiologia, Università Cattolica del Sacro Cuore, 00168 Roma, Italy; nikoladino.capocchiano@unicatt.it (N.D.C.); gianluca.franceschini@policlinicogemelli.it (G.F.); riccardo.masetti@policlinicogemelli.it (R.M.); 7Fondazione Policlinico Universitario “A. Gemelli” IRCCS, UOC di Chirurgia Senologica, Dipartimento di Scienze della Salute della Donna e del Bambino e di Sanità Pubblica, 00168 Roma, Italy; alba.dileone@policlinicogemelli.it (A.D.L.); stefano.magno@policlinicogemelli.it (S.M.); francesca.moschella@policlinicogemelli.it (F.M.); 8Fondazione Policlinico Universitario “A. Gemelli” IRCCS, UOC di Anatomia Patologica, Dipartimento di Scienze della Salute della Donna e del Bambino e di Sanità Pubblica, 00168 Roma, Italy; antonino.mule@policlinicogemelli.it (A.M.); danielaandreina.terribile@policlinicogemelli.it (D.A.T.); 9Department of Radiation Oncology, Iridium Kankernetwerk, 2170 Wilrijk-Antwerp, Belgium; philip.poortmans@telenet.be; 10Faculty of Medicine and Health Sciences, University of Antwerp, 2170 Wilrijk-Antwerp, Belgium; 11Radiation Oncology Section, University of Perugia and Perugia General Hospital, 06123 Perugia, Italy; cynthia.aristei@unipg.it

**Keywords:** breast cancer, large database, standardized data collection, networks

## Abstract

**Background:** During the 2016 Assisi Think Tank Meeting (ATTM) on breast cancer, the panel of experts proposed developing a validated system, based on rapid learning health care (RLHC) principles, to standardize inter-center data collection and promote personalized treatments for breast cancer. **Material and Methods:** The seven-step *Breast LArge DatabasE (BLADE)* project included data collection, analysis, application, and evaluation on a data-sharing platform. The multidisciplinary team developed a consensus-based ontology of validated variables with over 80% agreement. This English-language ontology constituted a breast cancer library with seven knowledge domains: baseline, primary systemic therapy, surgery, adjuvant systemic therapies, radiation therapy, follow-up, and toxicity. The library was uploaded to the *BLADE* domain. The safety of data encryption and preservation was tested according to General Data Protection Regulation (GDPR) guidelines on data from 15 clinical charts. The system was validated on 64 patients who had undergone post-mastectomy radiation therapy. In October 2018, the *BLADE* system was approved by the Ethical Committee of Fondazione Policlinico Gemelli IRCCS, Rome, Italy (Protocol No. 0043996/18). **Results:** From June 2016 to July 2019, the multidisciplinary team completed the work plan. An ontology of 218 validated variables was uploaded to the *BLADE* domain. The GDPR safety test confirmed encryption and data preservation (on 5000 random cases). All validation benchmarks were met. **Conclusion:**
*BLADE* is a support system for follow-up and assessment of breast cancer care. To successfully develop and validate it as the first standardized data collection system, multidisciplinary collaboration was crucial in selecting its ontology and knowledge domains. *BLADE* is suitable for multi-center uploading of retrospective and prospective clinical data, as it ensures anonymity and data privacy.

## 1. Introduction

Breast cancer, one of the main causes of women’s mortality, is characterized by highly complex presentation patterns [1]. Even though population-based screening programs [1], new therapies [2], advanced technologies [3], and multidisciplinary approaches [4] have improved survival and quality of life [4] in the previous decades, cure remains a challenge in some sub-groups of patients. Consequently, hypothesis-based tailored treatments that are adapted to each individual patient’s specific features are being explored in an approach termed personalized medicine. Due to complex information systems, personalized medicine overcomes uncertainties about particular conditions in small sub-groups of patients, which increase the complexity of decision-making [5,6]. Despite growing interest, a literature review revealed no consensus on how to define and apply personalized medicine [5]. Semantic approaches include patient stratification and treatment tailoring. In the former, individual patients are grouped into subpopulations according to the probability that a specific drug or treatment regimen will be of benefit, whereas in the latter, the individual patient’s status is used as the rationale for treatment choice [6,7].

The application of personalized medicine may be limited in clinical practice by the results of randomized controlled trials (RCTs). Patient selection, as defined by inclusion and exclusion criteria, leads to adaptive randomization, so outcomes refer only to the RCT-eligible population [8]. Furthermore, since the selected patients are usually in good clinical condition, with few or no comorbidities, the results cannot be extrapolated to all cases that physicians may encounter in clinical practice [9]. Additionally, due to long recruitment and follow-up times, RCT evidence may be out-of-date when it is made available, and progress may have already been made in developing treatments beyond old standards. Lambin et al. [10,11] reported that high quantity, low quality data from clinical charts reflected reality better than RCT data, and therefore provided valuable information for applying personalized medicine in clinical practice [9,12]. However, new instruments are needed to include the data and address uncertainties in clinical decision-making. 

Rapid learning health care fills this gap, since it extracts and applies knowledge from routine clinical care data rather than RCT evidence alone. Since data management of cross-linked information from diverse sources is complex, data analysis should be managed by machine learning to create decision support systems, i.e., software applications that apply knowledge-driven healthcare to clinical practice. Another rapid learning principle is that these systems need constant updating. 

In February 2016, a group of expert radiation oncologists organised the Assisi Think Tank Meeting (ATTM) to discuss research, controversies, and grey areas in breast cancer [13], and proposed a validated system based on rapid learning health care for standardized data collection to generate evidence for personalized medicine. In one of the participating centers, the Fondazione Policlinico Gemelli IRCCS, an umbrella protocol [14,15] was already approved by the Ethical Committee. The Beyond Ontology Awareness (BOA) platform (Figure 1) had been developed and implemented in close collaboration with physicians and informatics technology researchers [8,13]. It safely stores, analyzes, and shares data on diverse cancer types in a standardized manner [9,16] as well as reproducing the ontology structure and managing data legacy and privacy. BOA software converts the center’s legacy archives in accordance with a global data dictionary and anonymously replicates clinical data in a large cloud-based database. 

In the present project, the BOA platform was expanded for specific use in breast cancer care. A multi-disciplinary panel of experts from the Fondazione Policlinico Gemelli IRCCS, Perugia University, and General Hospital designed a standardized data collection system and developed the *Breast LArge DatabasE (BLADE)*. Its primary objective was to offer radiation oncology centers worldwide treating breast cancer the opportunity to collect and share data in a standardized large database, and thus develop descriptive, predictive, and prognostic models for supportive care, survival, and toxicity. Its long-term aim is to build decision support systems to personalize treatments, use resources in terms of cost-effectiveness, and make therapies more effective and less toxic.

## 2. Materials and Methods

After a review of breast cancer literature and current guidelines, a multi-step process was set up for data collection, analysis, application, and evaluation. Benchmarks were the rapid learning criteria by Lambin et al. [11]. The project was organized in a 7-step working plan as defined in a GANNT chart, and the time-frame for each step was established [17] (Figure 2). Data collection was structured to capture volume, variety, velocity, and veracity [11] and aimed to achieve a standardized ontology and overcome privacy issues. Approval was acquired from the Ethical Committee.

### 2.1. Data Collection Methodology

*Working Plan and Team (Step 1).* Members of the working group from the Fondazione Policlinico Gemelli and Perugia University, and General Hospital included 6 radiation oncologists; 1 medical oncologist; 1 pathologist; 3 breast surgeons; 1 radiologist; 2 informatics experts; 1 data manager. The working group established a timeframe of 12 months for developing the *BLADE* system. Responsibilities and times to complete each step were defined. Progress was updated every 3 months via live meetings or conference calls. 

*Variable Selection and Organization (Step 2).* Each team member reviewed the literature, focusing on RCTs and international guidelines, e.g., NCCN, ASTRO, ESTRO, and AIRO for radiation oncology [18,19,20] and established 7 domains of knowledge: baseline, primary systemic therapy, surgery, adjuvant systemic therapies, radiation therapy, follow-up, and toxicity. Major variables were chosen for each domain to create a shared-language ontology (terminology system). Variables were related to patients (e.g., age, sex, and gene profiling), clinical presentations (e.g., disease stage, markers, and pathology findings), treatments (e.g., surgery, systemic therapies, radiation therapy, and palliative care), and imaging (at diagnosis, treatment, and follow-up). 

Variables were validated by a consensus panel that indicated the response type for each variable (yes/no, single, or multi-options), selected and voted on multi-options. Consensus was reached with 80% agreement. 

*Setting up the BLADE domain (Step 3).* BOA was configured to include *BLADE* and process breast cancer data. It is equipped with local and cloud servers (Figure 1**)** depending on the desired configuration package. Users can access the BOA services through an intranet or internet connection and need only a standard web browser to connect, with no additional software. In the BOA.Local configuration, which only allows access through the local intranet, each institution has complete control over its data repository, and collected records are saved without any automated pseudo-anonymization procedures. The internet-facing server installation on the BOA.Cloud has the same features as the BOA.Local service, but it automatically and mandatorily pseudo-anonymizes all data. Before storage, each patient is assigned an ad hoc universally unique identifier (UUID), and all personal data or connections to existing records are severed. BOA.Cloud and BOA.Local store and process data in accordance with General Data Protection Regulation (GDPR). BOA.Local data can be dynamically cloned, automatically anonymized, and consolidated onto the BOA.Cloud server through a research manager—research node connection algorithm, and the data are then ready to be processed or analysed as needed. Figure 3 illustrates the underlying data model used in the databases of both BOA services. 

To create the *BLADE* domain, Excel spreadsheet files with all ontology-related variables were uploaded on to the BOA platform. *BLADE*’s 7 specific case report forms (CRFs), which were devised according to OpenClinica system criteria [21], are compatible with the BOA ontology framework. CRFs are available in Appendix A, with explanations of CRF definitions in Appendix A.

*Inclusion Criteria (Step 4).* The working group defined patient selection criteria, agreeing that retrospective and prospective data from all selected breast cancer patients can be included in *BLADE.*


*Retrospective data*: When *BLADE* is installed on the BOA platform, patient data will be derived from existing retrospective electronic or paper databases in each participating center. The data will be anonymized and shared only for research purposes. 

*Prospective data*: Patients whose data are eligible for enrolment in prospective *BLADE* studies will be informed about the opportunity to share their data for research purposes at their first medical examination, and invited to participate. The patients’ written informed consent will be obtained and archived.

*Patients’ privacy protection (Step 5).* Privacy needs to be guaranteed according to GDPR guidelines [22] for data protection. *BLADE* and BOA manage data using an AES-256 encryption system and an automatic data pseudo-anonymization algorithm. Each case is associated with a UUID code number with no reference to the individual’s identity, and is only accessible to specifically authorized health operators through their personal access codes and accounts. All changes in *BLADE* are automatically tracked and logged, including past and present values for form fields and the account identifiers of operators that modified existing data or inserted new data into CRFs. 

### 2.2. Testing the BLADE Domain for Coherency and Reliability (Step 6)

A data entry expert in the CRF system inserted data from 15 clinical charts of breast cancer patients that were randomly selected from Policlinico Gemelli records. According to GDPR principles, informatics verified accuracy, data conservation, limitations, and integrity during uploading. Criteria for coherency and reliability tests of the *BLADE* domain were the following (Article. 32 of GDPR):
−Data pseudo-anonymization and encryption;−Permanent assurance of confidentiality, integrity, availability, and resiliency of treatment systems and services;−Prompt restoration of availability and access to personal data in case of physical or technical accident;−Regular tests, verifications, and assessments of technical and organizational effectiveness measures to ensure data safety.

### 2.3. System Validation (Step 7)

*BLADE* was validated after checking adhesion to the GDPR criteria, and uploading and extracting data for statistical analysis from the clinical records of 64 patients who had undergone post-mastectomy radiation therapy (RT). All patients gave permission for their data from local databases to be transferred to *BLADE*.

Physicians asked the informatics expert to extract the following data from *BLADE*: Clinical-, treatment-, and tumor-related data: age, date of diagnosis, primary systemic treatments, histological sub-type, receptor status, multi-focality, and clinical and pathological stages;Reconstruction data: type of reconstructive surgery, prosthesis material, time to prosthesis-related complication (TPC), time to prosthesis reoperation (TPR), and ratio of TPC/time from reconstructive surgery;Dosimetric data referring to the chest wall: prescribed dose, conformity index, homogeneity index, and V95% and V105%.

Records were automatically extracted and the output was structured according to the standard needs of a data science team (e.g., a .csv file with all selected records processed on a flat table with specific column names and without any identifying information). 

Validation benchmarks were: −Uploading at least 80% of chart data by the data manager without physician assistance;−Physician correction of <20% of uploaded data;−Extraction of at least 80% of data for statistical analyses;−Joint physician and statistician correction of <20% of extracted data;−Performance of at least 80% of planned statistical analyses on RStudio^©^.

## 3. Results

### 3.1. Setting up BLADE (June 2016)

The 12-month timeline for completing *BLADE* overran by more than 1 year due to the quantity and complexity of the information. For example, Step 2 lasted 18 months, during which the working group met three times for variable selection and three times for variable validation. In July 2018, after reaching 80% consensus, a total of 218 variables were successfully uploaded to constitute the *BLADE* domain. Figure 4 reports as an example, the definition of the radiotherapy variable according to OpenClinica criteria.

The variables were organized into seven main CRFs corresponding to the knowledge domains, which were the interfaces for uploading encrypted patient data. In parallel with the data entry expert’s work, automatic testing tools in BOA tested specific characteristics in reference to the *BLADE* domain and generated synthetic patients. BOA tested both itself and the linked infrastructure by generating 5000 synthetic patients with a variable number of CRFs, and randomly created data in the space of nearly 20 min. To test performance, 30 fake user agents were connected to the interface and random pages from the web-service were requested for deletion or modification. Numbers for testing tool input were over a hypothetical maximum simultaneous workload for the *BLADE* project. Throughout these tests, no noticeable performance degradations were revealed, no abnormalities in the data structure or integrity were found, and no information leaked in the fake user sessions due to, for example, wrongly configured page-caching settings.

The privacy protection protocol was initially approved by the Ethical Committee of Fondazione Policlinico Gemelli IRCCS with protocol no. 0043996/18 in October 2018.

### 3.2. BLADE Data Safety Tests (January 2019)

To check that the *BLADE* domain was uploaded correctly, informatics analyzed accuracy, conservation limitation of data, data integrity, and data flows between application and data processing on 15 charts from randomly chosen patients. They completely adhered to EU GDPR criteria as reported in Article 32 Security of Processing [22,23]. Uploaded data were not linked to individual patients. Technical and organizational effectiveness measures did not break confidentiality, integrity, availability, and resiliency. Simulated physical and technical accidents showed no loss of data. 

### 3.3. Validation (February–July 2019)

The physician’s review increased 81.5% of uploaded data from 64 patients to 84% and corrected 10% of uploaded and missing data. The following were corrected: compile-time errors due to the data manager’s lack of experience with *BLADE* (7.5%); missing data (8.5%).

For statistical analysis, 100% of clinical, treatment and tumor-related data, 80% of reconstruction data, and 98% of dosimetric data were available. Mean available data ranged from 92.6% to 94.5%, corresponding to <20% validation benchmarks. All the planned statistical analyses were performed.

## 4. Discussion

The *BLADE* project was set up to support ATTM research into breast cancer, with the aim of providing decision support systems to facilitate clinical decision-making and treatment tailoring. In the 2016 ATTM [13], attention focused on developing such a system from the potentially large database that was available from clinical records, not only in radiation oncology centers, but in many other specialty units (e.g., surgery, pathology, medical oncology, etc.) that are dedicated to the diagnosis and treatment of breast cancer. 

The present results showed that *BLADE* is a valid system for collecting data anonymously, as its encryption system successfully passed the tests, satisfying GDPR criteria and benchmarks. Data managers were accountable for only 7.5% of errors, some of which were corrected during the physician’s review. Regarding radiation therapy, *BLADE* uniquely focuses on clinical, technical and dosimetric parameters, which makes it particularly suitable for analyzing radiation-therapy-related outcomes and toxicity. 

One of the strengths of *BLADE*’s ontology lies in its validated variables that were uploaded after a multi-step process involving the consensus of a multidisciplinary team. Unlike other large databases for breast cancer, *BLADE* provides health workers with the opportunity to focus on diverse fields in the diagnosis and treatment of breast cancer, as it is based on the acquisition of the pathways and the heterogeneity characterizing breast cancer [24,25,26,27,28]. Although several large national databases were set up, none were based on validated, published ontologies [25,26,27,28], and few could offer decision support systems [29,30,31,32,33]. Most were developed to investigate long-term survival outcomes such as, for example, the Surveillance, Epidemiology, and End Results (SEER) database, which was set up by the U.S. National Cancer Institute (NCI) and reports annually on the data it has collected on breast cancer from nine American oncological centers [29,30,31]. 

Another strength of the *BLADE* system is its capacity to incorporate new, validated variables or mathematical algorithms for assessing, for example, the success of treatment or a strategy for monitoring clinical outcomes and cost-effectiveness. In the future, it might include accreditation or valuation indicators for associated centers, update evidence or guidelines, and incorporate new sectors such as proteomics, complementary medicines, etc. 

One limiting factor of the present study was linked to *BLADE*’s small homogeneous sample and its inability to upload digital imaging and communications in medicine (DICOM) data, which will be very relevant when *BLADE* is used to develop prediction models. DICOM data and RT planning information will be uploaded with the 2022 *BLADE* upgrade, which will create a unique data repository [34]. A lack of testing of *BLADE*’s ability to perform machine learning analysis, an upcoming modality in cancer care, especially for predicting response to treatment, is a current limitation that is expected to be eliminated in the future. Using algorithms that iteratively learn from data, machine learning allows computers to find hidden insights without being explicitly programmed where to look, while inferential statistics need different tools to achieve this purpose, such as Bayesian networks, support vector machines, neural networking, and Cox regression. Machine learning is now starting to flank inferential statistical models (e.g., linear models, generalized linear models, and survival models), and its success over inferential statistics has already been reported together with the first promising results of its use in building predictive models of cancer survival [10,15,19]. We are confident that when *BLADE* is expanded to systematic multi-center use, machine learning analysis will become a reality and systems for decision-making support will be developed and validated, as *BLADE* is projected for a huge number of patients who will provide millions of data for analyses. 

In the near future, we will use *BLADE* in our clinical daily practice to collect retrospective and prospective data and analyze outcomes to assess the role of post-mastectomy radiation therapy in ductal in situ patients. This approach is derived from a 2019 survey by an ATTM research group [35], identifying this topic as a grey area in current practice. 

## 5. Conclusions

*BLADE*, one of the projects emerging from the 2016 ATTM [13], is a support system for breast cancer care. In successfully developing and validating it as a standardized data collection system, multidisciplinary collaboration was crucial for selecting its ontology and knowledge domains. *BLADE* is suitable for multi-center uploading of retrospective and prospective clinical data, as it ensures anonymity and data privacy. 

Finally, *BLADE* may constitute an international instrument for research purposes to be used by ATTM-like research groups [36].

## Figures and Tables

**Figure 1 jpm-11-00143-f001:**
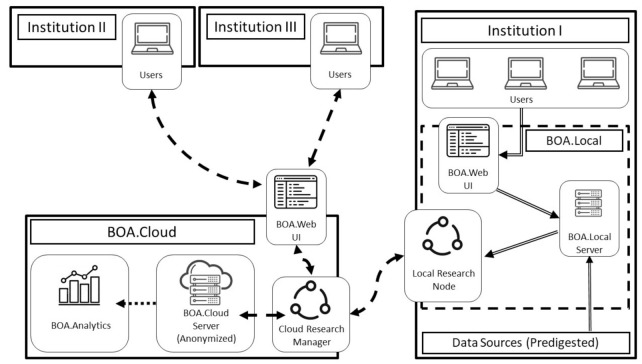
General beyond awareness ontology (BOA) architecture, with both the BOA.Local and BOA.Cloud servers. An infinite number of external institutions without a BOA.Local installation can be added at needed to this infrastructure. Double-line arrows represent non-anonymized patient data, dashed arrows represent anonymized patient data, and dotted arrows represent aggregate data.

**Figure 2 jpm-11-00143-f002:**
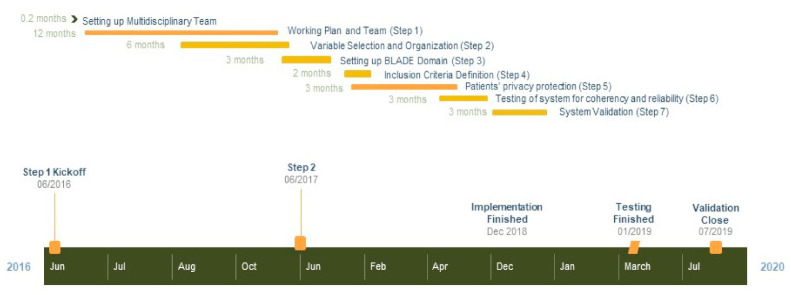
Timeline framework for ATTM.BLADE project.

**Figure 3 jpm-11-00143-f003:**
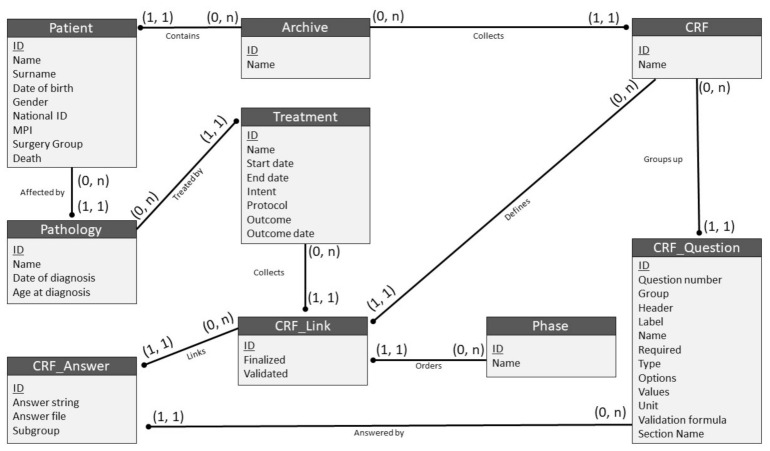
Underlying BOA data model visualized through an entity–relationship model that highlights all relationships between the different objects in the database. As an example (and using imaginary values), the archive named *BLADE* would contain a patient named John Doe, affected by a pathology of breast cancer, for which he was treated through a treatment of first treatment. This treatment would have a compiled version of the case report form (CRF) radiotherapy linked to the phase called neoadjuvant, and an answer of prone, to the question of radiotherapy treatment position present in the previously mentioned CRF.

**Figure 4 jpm-11-00143-f004:**
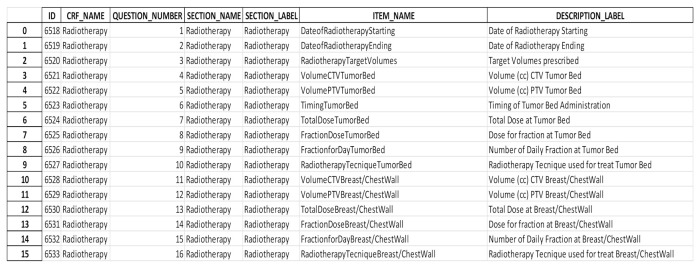
Example of a CRF configuration file. The columns represent various mandatory configuration settings for BOA and are to be interpreted as follows: The ID column represents an internal identifier and is generated automatically when the CRF is first uploaded. CRF_NAME refers to the name by which the CRF is to be visualized in the UI. QUESTION_NUMBER can either be automatically assigned or manually set, and refers to the ordering of the various questions inside the CRF, with SECTION_NAME and SECTION_LABEL working as visual dividers when the questions are displayed in the interface, with the former being the name to be used in the UI code, and the latter being the name to be displayed. ITEM_NAME and DESCRIPTION_LABEL work in a similar manner, with the former being the identifier in the underlying code and the latter being the name of the text to be displayed with the question in the UI.

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
