# Peer review of "The Assisi Think Tank Meeting Breast Large Database for Standardized Data Collection in Breast Cancer—ATTM.BLADE"

_jpm, 2021, doi:10.3390/jpm11020143_

Round 1
Reviewer 1 Report
This paper describes the process of setting up and validating a database system for breast cancer information aimed at relevant Radiation Oncology Centres. Although the process took longer than originally planned, the final result met GDPR and other validation criteria set. It is planned to be used to assess key identified clinical questions, and has the potential to be adapted to other settings.
The paper is clearly written with appropriate figures. It describes a useful system that could be used to advance breast cancer treatment. There are some minor comments listed below:
- Line 104 refers to “all Radiation Oncology Centres” – is this in a region, country, or worldwide?
- Could the authors comment more on how BLADE would be adaptable to different data collection / storage systems used by different institutions? Would BOA need to be in place?
- Some abbreviations were not defined (e.g. DICOM and RT).
- The author of reference 25 is incorrectly cited.
- There was no supplementary file 1 available.
- The supplementary file uploaded (file 2) still had tracked changes visible.
Author Response
This paper describes the process of setting up and validating a database system for breast cancer information aimed at relevant Radiation Oncology Centres. Although the process took longer than originally planned, the final result met GDPR and other validation criteria set. It is planned to be used to assess key identified clinical questions, and has the potential to be adapted to other settings.
We really thank the reviewer for his precious revision and comments and also for appreciating our efforts in system building.
The paper is clearly written with appropriate figures. It describes a useful system that could be used to advance breast cancer treatment. There are some minor comments listed below:
- Line 104 refers to “all Radiation Oncology Centres” – is this in a region, country, or worldwide? This definition is referred worldwide. The system was developed in English for facilitate international network.
- Could the authors comment more on how BLADE would be adaptable to different data collection / storage systems used by different institutions? Would BOA need to be in place? Thank you for question, we have the possiblity to clarify some aspect of our project. BLADE is supported by a codified ontology that can be adapted to data available in hospital system. We recently published an informatic architecture (10.3390/jpm11020065) project to collect data from hospital system and store them inside a repository data.
- Some abbreviations were not defined (e.g. DICOM and RT). We correct abbreviations
- The author of reference 25 is incorrectly cited. We correct citation 25
- There was no supplementary file 1 available. We attach CRFs in the new revision as supplementary file 1
- The supplementary file uploaded (file 2) still had tracked changes visible. We modified supplementary file 2
Reviewer 2 Report
Timely paper on novel automated data entry system that has meaningful promise in organizing complex data systems and maintaining them. this is preliminary work. Was pleased to see emphasis on patient privacy.
Author Response
We really thank the reviewer for appreciating our work and efforts spent in system and privacy management.